# Emotional Variability Analysis Based I-Vector for Speaker Verification in Under-Stress Conditions

**Barlian Henryranu Prasetio** [1,*] **, Hiroki Tamura** [2] **and Koichi Tanno** [2]

[1]   Interdisciplinary Graduate School of Agriculture and Engineering, University of Miyazaki, Miyazaki 889-2192, Japan

[2]   Faculty of Engineering, University of Miyazaki, Miyazaki 889-2192, Japan; htamura@cc.miyazaki-u.ac.jp (H.T.); tanno@cc.miyazaki-u.ac.jp (K.T.)

*   Correspondence: barlian@ub.ac.id

**Abstract:** Emotional conditions cause changes in the speech production system. It produces the differences in the acoustical characteristics compared to neutral conditions. The presence of emotion makes the performance of a speaker verification system degrade. In this paper, we propose a speaker modeling that accommodates the presence of emotions on the speech segments by extracting a speaker representation compactly. The speaker model is estimated by following a similar procedure to the i-vector technique, but it considerate the emotional effect as the channel variability component. We named this method as the emotional variability analysis (EVA). EVA represents the emotion subspace separately to the speaker subspace, like the joint factor analysis (JFA) model. The effectiveness of the proposed system is evaluated by comparing it with the standard i-vector system in the speaker verification task of the Speech Under Simulated and Actual Stress (SUSAS) dataset with three different scoring methods. The evaluation focus in terms of the equal error rate (EER). In addition, we also conducted an ablation study for a more comprehensive analysis of the EVA-based i-vector. Based on experiment results, the proposed system outperformed the standard i-vector system and achieved state-of-the-art results in the verification task for the under-stressed speakers.

**Keywords:** speaker verification; emotional conditions; stress speech; eigenemotion; i-vector technique; joint factor analysis

## 1. Introduction

Speaker verification is the process of accepting or rejecting the identity claim of a speaker [1]. This system is commonly used for the applications that use the voice as the identity confirmation, known as biometrics, natural language technologies [2] or as a pre-processing part of the speaker-dependent system, such as conversational-based algorithms [3,4]. Many methods have been explored in terms of verification task [5]. Most of them has proven their robustness in the background noises. However, just a little work that observed the effects of the emotional conditions on the speech characteristics. However, emotional conditions (especially stress conditions) are the most crucial factor that highly impacted the voice tone's characteristics.

Stress is one of the unconscious emotions [6] that occurs due to environmental stimuli [7]. Everyone expresses stress in different ways. It depends on their individual related to experience background, and emotional tendencies [8]. Physically, as a response to stress, the body releases certain hormones that could increase the rhythm of heart and breathing rate and muscle tension [9]. It makes the respiratory system and timing of the vocal system physiology changed [10], and the fundamental characteristics of voice tone also changed [11]. This condition causes the performance of

the speaker-based systems, such as the speaker verification system, generally decreases [12]. Therefore, the presence of emotions should be accommodated to obtain an accurate speaker verification result.

To this end, the works of [13,14] identified emotions and speakers in separate tasks. Some studies proposed to eliminate the effect of emotion by applying channel compensation strategy [15] or emotion-dependent score normalization [11]. It indicates that the most speaker verification system assumed emotions are the negative effect that should be alleviated due to mismatch conditions between training and testing or real condition. Since this paper is to be addressed as a pre-processing part of the speaker-dependent system [3,4], we take action to be in line with emotion variability so that it is able to verify the speakers even though under-stress conditions.

In this decade, a robust and effective technique for recognizing the speaker has been introduced by [16,17], known as the identity vector (i-vector). I-vector is a statistical model that is collected from a trained Gaussian mixture model (GMM) to represent a Universal Background Model (UBM). In contrast to the joint factor analysis (JFA) that model the speaker and channel variability of GMM supervector separately, i-vector models both variabilities in a single low-dimensional space using the total-variability model (TVM). Thus, the efficiency of i-vector representation is dealing with low-dimensional vectors rather than with the high-dimensional space of the GMM supervectors. Despite effective due to its low-dimensional representation, TVM does not model the speaker and channel subspace as well as the JFA model.

Thus, we propose to take advantage of the JFA model, but it also accommodates the presence of emotions (especially stress) in the speaker verification system. Since the emotion effect is similar to the channel effect [15], we explicitly use the channel variability component of the JFA model as the emotional variability component. We named this method as the emotional variability analysis (EVA). Then, the i-vectors technique is performed to exploit the possibility of representing the speaker vectors in a low-dimensional space, called EVA-based i-vector.

The advantage of the proposed idea is to rely on the eigenchannel space, which accommodates the emotional variability so that it has an ability to recognize the speaker in emotional conditions accurately than the total-variability model. The proposed system is addressed to verify the speaker in emotional stress conditions by estimating a linear transformation that allows keeping the span of the speaker-specific eigenemotion subspace, which it then provides better representation for the i-vector extractor. We consider each training segment as belonging to a different speaker, as standard i-vector's training procedure by applying the minimum divergence estimation (MDE) during the training iterations for obtaining a new variability matrix. Finally, the proposed method contributes to development an effective speaker model without worrying about the emotional conditions of the speaker.

The rest of this paper is organized as follows: Section 2 discusses the existing speaker verification methods and its related works. Section 3 introduces the proposed speaker verification system, the compensation method, and the scoring algorithm. The experimental setup that is consists of the use of dataset and parameters setting of the proposed and baseline system are presented in Section 4. The experiment and their result are discussed in Section 5. Section 6 depicts the conclusions and future work.

## 2. Related Works

The performance of the speaker verification system is affected by many factors. It could come from an external and internal source. Both sources bring a negative effect on the system by presenting extra vocal variability. Emotion is one of the internal sources that most affects the performance of the speaker verification system. Emotion induce intra-speaker vocal variability [18], even more for stress condition [19].

Many feature extraction technique has been explored their effectiveness in emotional speaker verification systems, such as Mel-frequency cepstral coefficients (MFCC) [20–23] and Linear Predictive Cepstral Coefficients (LPCC) [21]. Both techniques might achieve a good result in recognizing the

speaker, but they were not directly used to address the speaker-specific emotional information. In order to handle the presence of emotion, the other studies decided to identify the emotion first and verify the speaker in the next step [13,14]. In practice, emotion discrepancy between training and evaluation provoke a problem on the channel effect. Therefore, [24,25] proposed an adaptation approach that employs a model or feature transformation. Due to the speaker verification system aimed to verify whether an input speech corresponds to a claimed identity, many works assumed that emotion is a negative effect so that intra-speaker emotion variability should be alleviated. To this end, [15] applied the emotion compensation method, which is expected to be capable of removing emotion variability. Another approach performed a score normalization to estimate emotion-dependent biases, and then remove them from verification scores [11].

On the other hand, JFA and i-vector have spent many efforts to end the problems of channel variability. In this decades, the JFA model has become a state-of-the-art approach in many speech-related systems, including the speaker recognition task. It is related to the speaker and channel factors of the GMM supervector that are represented comprehensively. Since there are some correlations between the eigenchannel and the eigenvoice factors that bring the information about the speaker identity, the speaker and channel variability are modeled in a single low-dimensional space by TVM, known as the i-vector model.

The JFA and i-vector model has been widely used in many speech-based systems and showed an extreme ability in the highly accurate result. However, both approaches were not designed explicitly for emotional conditions. Moreover, a few studies have revealed that there is an emotion variability issue. A study [11] mentioned there is a similarity between channel and emotion effect. Ref. [26] explicitly assumed emotion variability as a special kind of channel variability that predictable under each emotional state as under each channel.

In this paper, we propose a speaker verification model by borrowing the idea from the JFA approach and i-vector technique, termed as the emotional variability analysis (EVA)-based i-vector. EVA leverage a channel component of the JFA model to accommodate emotional variability. The span of the eigenemotion and eigenvoice subspace in further is represented in a low-dimensional space using the i-vector technique. Unlike EFA approach [26] that define the speaker and emotion in a single supervector representation, EVA present speaker and emotion in a different component, eigenvoice, and eigenemotion. Each component is trained using a standard i-vector procedure for obtaining a new variability matrix.

## 3. Materials and Method

The proposed speaker verification system consists of three phases: development, enrollment, and verification phase, as shown in Figure 1. The development phase is the process of learning speaker-independent parameters. A set of labeled speaker and emotion of the Speech Under Simulated and Actual Stress (SUSAS) database is used to train the GMM-UBM framework. Then, the speaker GMM supervector is modeled using the EVA algorithm that decomposed into the speaker, emotion, and residual subspace. Then, the EVA is modeled into a low-dimensional total-variability space, called the EVA-based i-vector model. The total-variability model is trained using a similar process of EVA. In the enrollment phase, the speaker's feature is extracted using the EVA-based i-vector technique. The channel compensation method (deep discriminant analysis or DDA) is used to generate speaker models using the EVA-based i-vector features as input, then registered it to the database. In the verification phase, the same procedure is conducted to the verification data. The verification data is a conversation data that consists of sequence utterances with an unknown speaker (unlabeled). For verification, we assume the first utterance is spoken by speaker-1, identified into the registered speaker models. Then, we verify whether the second utterance is also spoken by speaker-1 using the similarity algorithm. Finally, the acceptance result is decided based on the equal error rate (EER) in terms of the decision threshold.

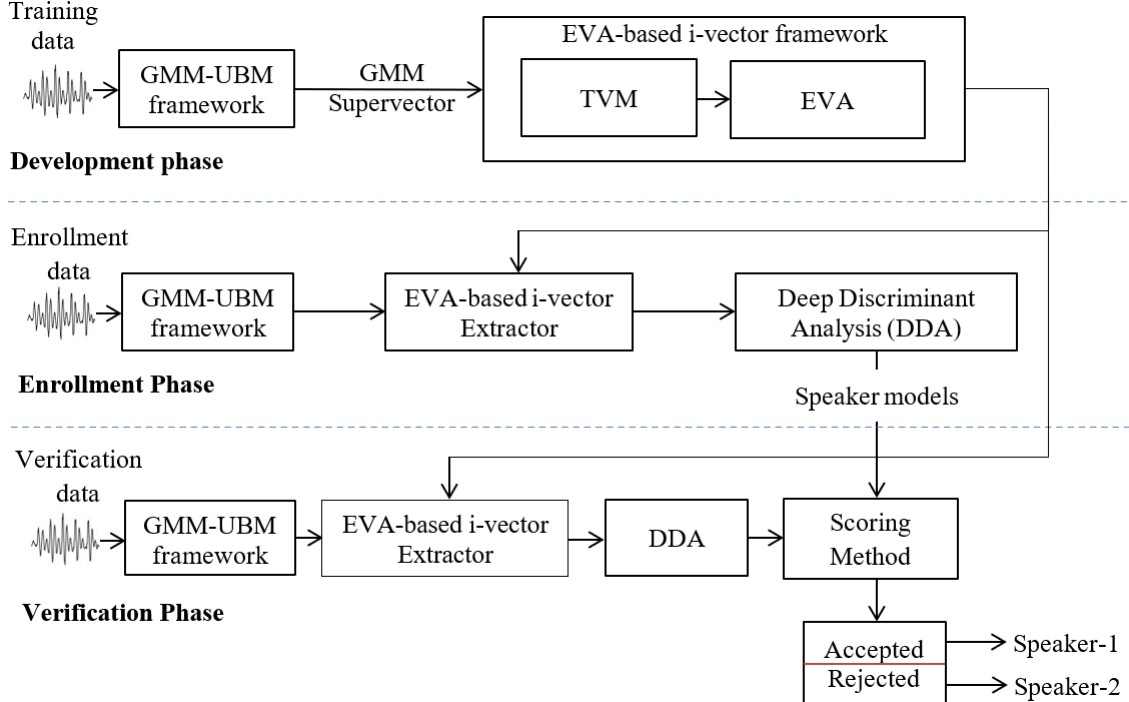

**Figure 1.** The proposed speaker verification system in development, enrollment, and verification phase.

### 3.1. EVA-Based I-Vector Framework

GMM/UBM supervector composed of the speaker-independent, speaker-dependent, channel-dependent, and residuals components [17,27]. Each component is represented in a set of low-dimensional factors (eigen-dimensions) of the corresponding component by JFA [28]. Thus, a speaker supervector *s* composed of three subspaces: speech subspace (eigenvoice matrix *V*) and channels subspace (eigenchannel matrix *U*), and residuals subspace *D*, formulated as follows:

$$s = m + Vy + Ux + Dz \tag{1}$$

where *m* is UBM supervector, *x* is the channel factors, *y* is speaker factors, and *z* is the speaker-specific residual factors.

Since there are some eigenvoice factors that are maintained by eigenchannel factors [17], all variability components can be modeled into a single low-dimensional variability component, known as the total-variability model (TVM). I-vectors use the total-variability factor *w* to represents each speech sample. Each factor controls an eigen-dimension of the total-variability matrix *T*, expressed as follows:

$$s = m + Tw \tag{2}$$

Although the *T* matrix has been spanned for all variability components, the *T* matrix could not represent the speaker and channel subspace as well as the JFA model, especially for speaker under-emotional conditions. As mentioned in Section 1, due to the emotional effect is similar to the channel effect, we take advantage to use eigenchannel subspace of the JFA model as the eigenemotion subspace. Hence, we propose to present a more informative speaker supervector, which accommodate emotional variability by leveraging the JFA, known as emotional variability analysis (EVA). We keep using the i-vectors framework to exploit the possibility of representing a speech segment in a low-dimensional space. We estimate a different *T* matrix composition, which better accounts for the speaker in emotional condition. EVA is similar to the JFA, but it is estimated with the emotional constraint that it spans the same subspace represented by the eigenchannel matrix

trained on the same dataset. EVA defines a given speaker supervector $s$ in a similar form of JFA model (Equation (1)) that could be decomposed as follows:

$$s = m + Vy + Ex + Dz \tag{3}$$

where $E$ is emotion subspace (eigenemotion matrix) and $x$ is the emotional factors.

### 3.1.1. Subspace Estimation

As described in [29], we train the EVA matrices as follow:

1. The eigenvoice matrix $V$ is trained by assuming $E$ and $D$ are zero.
2. The eigenemotion matrix $E$ is trained using a given estimate of $V$, and by assuming $D$ is zero.
3. The residual matrix $D$ is trained using given estimates of $V$ and $E$.

Since the $V$ matrix has highlighted in obtaining the speaker-based principal dimensions, we use the speaker-subtracted statistics to train $E$ like the training process of $V$. In other words, the $E$ matrix focuses on obtaining the emotion-based principal dimensions. As described above, the $E$ matrix is trained using the estimated $V$ and assumes $D$ is zero. The 0th ($N_c$) and 1st ($F_c$) order statistics for each speech data ($spd$) of each speaker ($s$) in Gaussian mixture component ($c$) are expressed as follow:

$$
\begin{aligned}
N_c(spd, s) &= \sum_{t \in spd, s} \gamma_t(c) \\
F_c(spd, s) &= \sum_{t \in spd, s} \gamma_t(c) Y_t
\end{aligned}
\tag{4}
$$

where $\gamma_t(c)$ is the posterior of Gaussian component $c$ for observation $t$ of speaker $s$. Then, we compute the speaker shift using matrix $V$ and factor $y$.

$$spkshift(s) = m + V * y(s) \tag{5}$$

For each speech data of each speaker, we subtract Gaussian posterior-weighted speaker shift from the first-order statistics.

$$\hat{F}_c(spd, s) = F_c(spd, s) - spkshift(s) * N_c(spd, s) \tag{6}$$

We then expand the order statistics into matrices as follow:

$$
\begin{aligned}
NN(spd, s) &= \begin{bmatrix} N_1(spd, s) * I & & \\ & \ddots & \\ & & N_c(spd, s) * I \end{bmatrix} \\
FF(spd, s) &= \begin{bmatrix} \hat{F}_1(spd, s) \\ \vdots \\ \hat{F}_c(spd, s) \end{bmatrix}
\end{aligned}
\tag{7}
$$

where $I$ is identity matrix.

Then, both matrices $NN(spd, s)$ and $FF(spd, s)$ are used to train matrix $E$ and factor $x$. The estimate of matrix $E$ is defined as follows:

$$
E = \begin{bmatrix} E_1 \\ \vdots \\ E_c \end{bmatrix} = \begin{bmatrix} A_1^{-1} * \mathbb{C}_1 \\ \vdots \\ A_c^{-1} * \mathbb{C}_c \end{bmatrix} \quad \text{where} \quad \mathbb{C} = \begin{bmatrix} \mathbb{C}_1 \\ \vdots \\ \mathbb{C}_c \end{bmatrix}
\tag{8}
$$

where $A_c = \sum_s N_c(spd,s)l_e^{-1}(spd,s)$, $\mathbb{C} = \sum_s FF(spd,s) * (l_e^{-1}(spd,s) * E^* * \sum^{-1} *FF(spd,s))^*$, and $l_e$ is covariance of posterior distribution of $x(spd,s)$.

### 3.1.2. Linear Score Computation

By using the matrices ($V$,$E$, and $D$), we estimate the $y$ (speaker factors), $x$ (emotion factors), and $z$ (residual factors), in terms of their posterior means given the observation [30]. We then compute the final score using the matrices and factors. For the test speech data *test* and the target speaker speech data *target*, we obtain the final score via linear product as follows:

$$Score = (V * y(target) + D * z(target))^* * \sum^{-1} *(FF(test) - NN(test)) * m - NN(test) * E * x(test) \quad (9)$$

### 3.1.3. I-Vector Technique

We use the i-vector technique on the EVA algorithm to model the speaker. I-vector technique models the total-variability factor $w$ in a single low-rank space in representing each speech data, as expressed in Equation (2). We train matrix $T$ using the same procedure with the training of matrix $V$ in EVA but treat all speech data of all training speakers as belonging to different speakers. Finally, by given $T$, we obtain i-vectors $w$ for each speech data.

### 3.2. Deep Discriminant Analysis

In speaker recognition, the i-vector system is typically followed by linear discriminant analysis (LDA) for channel compensation method. LDA is widely used in pattern recognition tasks to project features into a lower-dimensional and more discriminative space. The simpleness of LDA makes it popular to reduce the vector dimensions and channel compensations for the i-vectors. LDA transform a linear representation of a high-dimensional feature vector (i-vector) $x$ into a low-dimensional discriminative subspace $y$ that projected as $W$ matrix, ($W : \mathbb{R}^h \to \mathbb{R}^l$), formulated as follows:

$$y = W^T(x) \quad (10)$$

where $W$ is a matrix that represent the inter-class $S_b$ and intra-class $S_w$ covariance matrix of the two classes that to be discriminated. The LDA defines the criteria $\theta$ to separate two classes, as follows:

$$\theta = \frac{W^T S_b W}{W^T S_w W} \quad (11)$$

The projection matrix $W$ contains the eigenvectors that correspond to the largest eigenvalue of $S_w^{-1} S_b$, chosen as a solution for LDA optimization.

By assuming all classes shared the same covariance matrix, LDA becomes a simple and effective compensation method in Gaussian distributed data. However, the presence of the emotional state could the speaker's factors fluctuated non-normally [31]. To address this issue, we use a discriminative DNN-based compensation method in the i-vector space, as proposed by [32]. This approach capabilities have proven as explored by [33–35]. Ref. [32] termed this compensation method as deep discriminant analysis (DDA).

DDA is a DNN-based compensation method that shares the same spirit with LDA. As typically DNN, DDA is trained using the stochastic gradient descent (SGD) optimization algorithm, and weights are updated using the backpropagation algorithm. In the context of an optimization algorithm, the error must be evaluated iteratively. It requires the choice of the loss function that can be used to estimate the loss of the model. The choice of loss function must match the framing of the specific predictive modeling problem, such as classification or regression. Mean squared error (MSE) and mean absolute error (MAE) are suitable for regression loss functions. For the classification tasks, the possible loss functions are cross-entropy loss and Kullback–Leibler (KL) Divergence Loss.

The cross-entropy loss is closely related to KL divergence, but it is different. KL divergence calculates the relative entropy between two probability distributions, whereas cross-entropy can be thought to calculate the total entropy between the distributions. Cross-entropy is also usually confused with logistic loss. It is because both losses calculate the same quantity when used for classification problems. Cross-entropy is a good cost function when works on classification tasks and uses SoftMax activation functions in the output layer that model probabilities distributions. This combination is often known as SoftMax loss. Another effective loss function that can improve the discriminative power of the deep learned features has been introduced, known as the center loss. Center loss is performed by minimizing the intra-class variations while keeping the features of different classes separable.

By joint supervision of SoftMax loss and center loss, the learned embeddings of different classes staying apart (SoftMax function), while the embedding from the same class are pulled close to the centers. The joint supervision of SoftMax loss $\mathcal{L}_s$ and center loss $\mathcal{L}_s$ of DDA is formulated as follows:

$$\mathcal{L} = \mathcal{L}_s + \lambda \mathcal{L}_c$$
$$\textit{where,}$$
$$\mathcal{L}_S = -\sum_{i=1}^{N} log \frac{e^{W_{s_i}^T x_i + b_{s_i}}}{\sum_{j=1}^{S} e^{W_j^T x_i + b_j}} \tag{12}$$
$$\mathcal{L}_C = \frac{1}{2} \sum_{i=1}^{N} \| x_i - c_{s_i} \|^2$$

where $\lambda$ is the balancing of both loss functions. The $x_i \in \mathbb{R}^d$ denotes $i$th sample, belonging to $s_{ith}$ class, and $S$ denotes the number of SoftMax outputs (number of classes). $W_j$ is $j$th column of the weight matrix $W$ and $b$ is bias term. $d$ and $N$ are feature dimensions and the total number of training samples (i-vector), respectively. The $c_{s_i}$ represents the $s_{ith}$ class center.

Structurally, DDA consists of four layers: an input layer, a hidden layer, an embedding layer, and a loss layer. A detailed network structure of DDA is shown in Table 1. The extracted i-vector from different speakers are used as input. The transformed features are extracted from the embedding layer. In the compensation stage, the source i-vectors are mapped to their corresponding transformed version through the trained neural network. Similar to the projection transformation of LDA (Equation (10)), DDA compensates for the original i-vector $x$ to lower-dimensional embedding space $y$, expressed as:

$$y = \mathcal{G}(x) \tag{13}$$

where $\mathcal{G}()$ denotes the non-linear transformation function of DDA.

**Table 1.** The network structure of the deep discriminant analysis.

| Layer | Number of Neurons | Non-Linear Function |
|-------|-------------------|---------------------|
| Input | 600 | ReLU |
| Hidden | 600 | ReLU+BatchNorm |
| Embedding | 400 | - |
| Loss | Softmax loss | $\lambda$*center loss |

### 3.3. Scoring Method

In the back-end part, the speaker verification system evaluates the similarity between two speakers based on their identity features. Since the output of a front-end part is inter-session and speaker-dependent, statistical modeling is employed as a compensation method, such as LDA. However, the compensation methods do not provide a final score of verification result. Therefore, the i-vector systems are typically followed by a scoring method after the compensation method.

The score of speaker verification is produced by comparing two i-vectors from different speakers. This score gives an estimation of the log-likelihood ratio between the same-speaker and different speaker hypotheses. Ref. [36] reported that the probabilistic linear discriminant analysis (PLDA) presents a better result than the cosine algorithm. Another study reported [32] that Cosine or Euclidean scoring methods provide a significant improvement than PLDA. The effectiveness of the Mahalanobis scoring method has been explored by [37,38] and presented an excellent performance for the i-vector system in the speaker recognition system. In this paper, we assess the effectiveness of the speaker verification system in different scoring methods, such as Cosine similarity scoring (CSS), Euclidean distance scoring (EDS), and Mahalanobis distance scoring (MDS).

In i-vector space, cosine similarity is a state-of-the-art scoring method in the speaker verification field. Although PLDA usually gives better accuracy, Cosine similarity scoring (CSS) remains a widely used method due to simplicity and acceptable performance. The cosine similarity between the target i-vector $w_{target}$ and the test i-vector $w_{test}$ is computed as follows:

$$Score_{CSS} = \frac{w_{target}^T w_{test}}{\parallel w_{target} \parallel \parallel w_{test} \parallel} \tag{14}$$

Euclidean distance is defined as the Minkowski distance of the 2-norm distance. In speaker verification, Euclidean distance scoring (EDS) is defined as the straight-line distance between two i-vectors, $w_{target}$ and $w_{test}$, defined as follows:

$$Score_{EDS} = \sqrt{\parallel w_{target} - w_{test} \parallel^2} \tag{15}$$

The effectiveness of the Mahalanobis metric for speaker detection scoring has been proven by [37,38]. The score between two i-vectors $w_{target}$ and $w_{test}$ is proportional to the log-probability that both i-vectors belong to a unique class following the covariance matrix $\tau$. The centroid of this hypothetical class could be $w_{target}$ or $w_{test}$ or their mean, knowing that each proposition gives equivalent results. The final Mahalanobis distance scoring (MDS) function is:

$$Score_{MDS} = - \parallel w_{target} - w_{test} \parallel_{\tau^{-1}}^2 \tag{16}$$

*3.4. Decision Evaluation Metrics*

We evaluate the proposed system using the equal error rate (EER) evaluation metrics for the verification tasks (accepted or rejected). The evaluation is obtained from two main errors: False Alarm Rate (FAR) and False Rejection Rate (FRR). EER is a intersect between FAR and FRR, or in other words, at which FAR and FRR are equal. We define an EER in terms of the decision threshold $\eta$, as follows:

$$EER = \frac{FRR(\eta) + FRR(\eta)}{2} \tag{17}$$

## 4. Experimental Setup

*4.1. Dataset*

We used the stress speech data from the Speech Under Simulated and Actual Stress (SUSAS) database that was collected by the Linguistic Data Consortium (LCD) [39]. The SUSAS database consisted of short utterances labeled data and the unlabeled conversation data [40]. We used the labeled data for development and enrollment which is consisted of seven speakers with five class labels. For the verification phase, we used six unlabeled conversation data that spoken by two speakers. The detailed information of the data used in the experiments is shown in Table 2. In the experiments, the unlabeled conversation data has been normalized using a speech activity detection (SAD) system [33].

**Table 2.** The dataset used in the experiments. (**a**) is the data used for development and enrollment phase while (**b**) is the data used for verification phase.

| (a) | |
| --- | --- |
| **Class** | **Number of Utterances** |
| High stress | 620 |
| Low stress | 620 |
| Neutral | 620 |
| Angry | 436 |
| Soft | 420 |

| (b) | |
| --- | --- |
| **Data ID** | **Number of Utterances** |
| 1 | 98 |
| 2 | 102 |
| 3 | 94 |
| 4 | 118 |
| 5 | 107 |
| 6 | 97 |

*4.2. Proposed System Setting*

The 13-dimensional acoustic feature vector was extracted using the MFCC technique with 25 ms frame length that was normalized for each sliding window. A 512-mixture GMM-UBM system is applied to generate a 600-dimensional EVA-based i-vector feature. The dimensions of each EVA component are as follows:

- Matrix V: 20,000 by 300 (300 eigenvoice components)
- Vector y: 300 by 1 (300 speaker factors)
- Matrix E: 20,000 by 100 (100 eigenemotion components)
- Vector x: 100 by 1 (100 emotion factors)
- Matrix D: 20,000 by 20,000 (20,000 residual components)
- Vector z: 20,000 by 1 (20,000 speaker-specific residual components)

In the back-end part, as shown in Table 1, DDA contains one input layer, one hidden layer, and one embedding layer. ReLU is used as an activation function, and we incorporated a batch normalization layer in the hidden layer to stabilize the training procedure. We set the weight balancing parameter for SoftMax loss and center loss $\lambda = 10^2$ and the controller parameter for the learning rate of center $\alpha = 10^1$. DDA is performed to reduce the EVA-based i-vector dimensions into 400 [32].

*4.3. Baseline System Setting*

We use a standard i-vector/DDA as a baseline system for the speaker verification task. A 600-dimensional i-vector feature is generated using the same setting with EVA-based i-vector (see Section 4.2). In the baseline system, DDA is performed to reduce the i-vector dimensions into 400.

**5. Result and Discussions**

In this section, we evaluate the effectiveness of the proposed system in verifying the speaker of the conversation data of the SUSAS dataset. We present the evaluation in terms of the equal error rate (EER). The performance comparison between the proposed speaker verification system and baseline methods are presented in Section 5.1. To demonstrate the further capabilities of EVA, we provide an ablation study and its advanced analysis in Section 5.2.

*5.1. Evaluation Result*

The proposed system is evaluated in the speaker verification task of the dataset described in Section 4.1. In this experiment, we set the proposed and baseline system described in Sections 4.2 and 4.3. As shown in Table 3, the proposed system presents a better result compared to the baseline system and obtains a significant improvement in all scoring methods. EDS assumes the data to be isotropically Gaussian, i.e., it would treat each feature equally. On the other hand, the MDS seeks to measure the correlation between variables by assuming an anisotropic Gaussian distribution instead. Since EVA compensates for the emotions, there are some correlations between emotion and speaker's supervector. Therefore, it is foreseeable that MDS outperforms the EDS empirically. Thus, the best performance of the baseline system (EER 6.19%) is achieved for the EDS scoring method, while MDS obtains the best performance for EVA-based i-vector with the EER of 4.08%.

**Table 3.** The speaker verification system performance in terms of % equal error rate (EER).

| Method | Scoring Method | | |
|---|---|---|---|
| | CSS | EDS | MDS |
| Baseline system | 6.78 | 6.19 | 7.63 |
| Proposed system | 4.51 | 4.37 | 4.08 |

To provide insight into the effectiveness of EVA-based i-vector at both the individual and population levels, we visualize the vector distribution of the speaker using the t-distributed Stochastic Neighbor Embedding (t-SNE). The t-SNE is a dimensionality reduction technique that maps similar objects to nearby points and dissimilar objects to distant points. The class distribution of standard i-vector and EVA-based i-vector are presented in Figure 2. As shown in Figure 2, EVA-based i-vector (Figure 2b) represents speaker vectors more separately, and it saw that some points gather in some sub-clusters area. It indicates that EVA-based i-vector can recognize the speaker effectively in different emotions by presenting a sub-cluster of emotion.

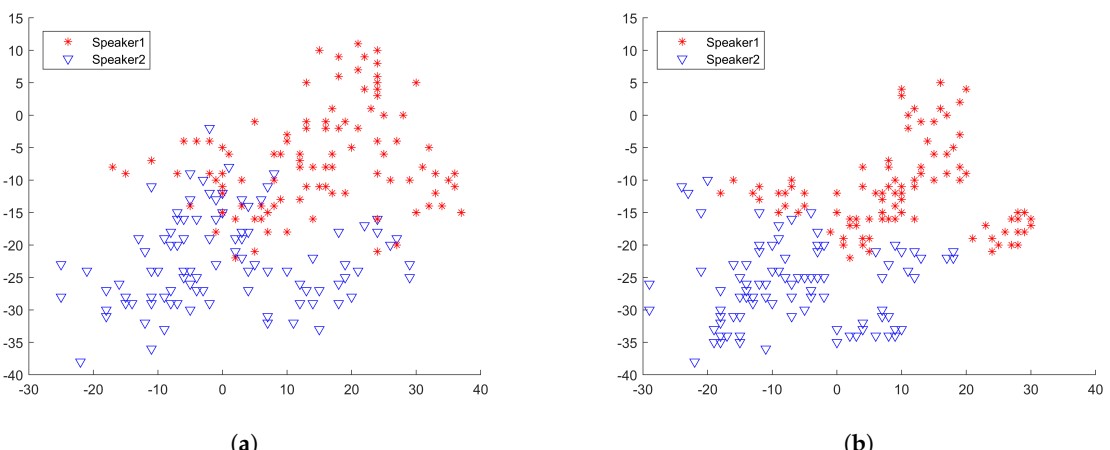

(**a**)　　　　　　　　　　　　　　　　　　　　　　　　　(**b**)

**Figure 2.** The t-SNE visualization of the speaker class distribution of the first conversation data. On each sub-figure, we visualize 100-feature vectors (generated randomly). (**a**) denote the class distribution for standard i-vector while (**b**) shows the class distribution for EVA-based i-vector.

*5.2. Ablation Experiment*

To analyze the proposed EVA in further, we conducted an ablation experiment to explore the effect of involved parameters on the verification result. We examine whether the parameter of the number of mixture GMM-UBM system and extracted feature dimensions affect the model performance. Therefore, it is possible to investigate the EVA reasonably quite well in terms of the equal error rate (EER).

### 5.2.1. The Effect of the Number of Mixture GMM-UBM System

EVA-based i-vector in which the acoustic features (MFCC and log-energy plus their 1st and 2nd derivatives) are generated by a GMM. GMM supervector representing the *i*th utterance is assumed to be generated by the factor analysis model of Equation (2), and could be written in a component-wise form:

$$s_{ic} = m_c + T_c w_i + \epsilon_{ic}, \quad c = 1...C \tag{18}$$

where $\epsilon_{ic}$ is the residual noise following a zero-mean Gaussian distribution. As shown in Equation (18), the speaker supervector is a function of the number of mixture GMM-UBM system. Thus, it is related to model performance. Therefore, we explore how far the number of mixtures the GMM-UBM system is related to model performance.

In this experiment, we set i-vector dimensions to 600 and reduce it into 400-dimensional by DDA. For scoring, we observe the system performance for different scoring algorithms, i.e., CSS, EDS, and MDS. Figure 3 shows the EER performance of the speaker verification system in the different number of mixtures GMM-UBM system. The increase in the number of mixture GMM-UBM components could significantly improve system performance. The lowest EER is achieved when the number of mixture GMM-UBM by 512 for all scoring algorithms. In this experiment, MDS shows its best performance as a scoring algorithm by presenting the lowest EER of 4.07%, and relatively more stable than others for the number of mixture GMM-UBM of 1024 and 2048.

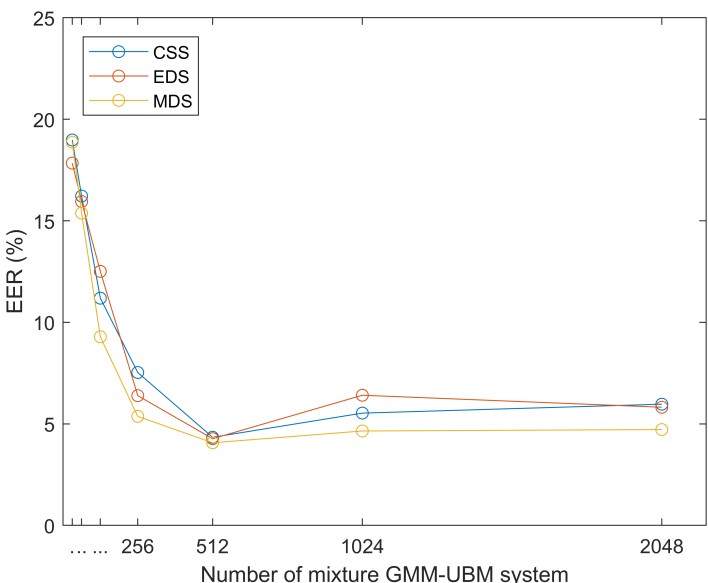

**Figure 3.** The EER performance of speaker verification system in different number of mixture GMM-UBM system of EVA.

### 5.2.2. The Effect of the Number of EVA-Based I-Vector Dimensions

The total-variability space is presented by a total-variability matrix $T$. As shown in Equation (2), $T$ is the low dimension square matrix as it is obtained from the selected Eigenvectors of the corresponding larger Eigenvalues of the total-variability space. The Eigenvector gives the direction along with the maximum variability in low-dimensional space. The matrix $T$ with the low ranks and $w$ is the random vector such that $w = \{w_i\}_{i\,1}^c$, $w$ follows the standard normal distribution. The components associated with each vector $w_i$ is the feature factors collected from the matrix $T$. Each vector of $\{w_i\}_{i\,1}^c$ follows the identical type distribution, known as "i-vector". Therefore, the extracted i-vectors contain both intra and inter-accent variations. Thus, the dimensionality of i-vector projects how optimal the accent variability is maximized, and intra-accent variability is minimized. Hence, we observe whether the dimension of EVA-based i-vector related to the speaker

verification results in three different i-vector dimensions, as presented in Table 4. We found that accuracy improves with the increase of i-vector dimensionality for all scoring methods. The low EER is given by 600-dimensional with 4.36%, 4.53%, and 4.11% respectively for CSS, EDS, and MDS. In further, MDS presents the lowest EER and the stablest scoring method. It is because MDS performed non-singular linear transformations so that it is not affected by feature dimensions.

**Table 4.** The EER comparison of speaker verification system in different extracted dimensions of EVA-based i-vector.

| Scoring Method | Dimensions | | |
|:---:|:---:|:---:|:---:|
| | **600** | **500** | **400** |
| CSS | 4.36 | 4.87 | 5.21 |
| EDS | 4.53 | 4.54 | 4.56 |
| MDS | 4.11 | 4.12 | 4.12 |

## 6. Conclusions

In this paper, we proposed an effective speaker verification system that accommodates the presence of the emotional conditions of the speaker. We addressed the proposed system as a pre-processing part of the speaker-dependent system. The proposed system used the emotional variability analysis (EVA)-based i-vector model as the feature extractor, and the deep discriminant analysis (DDA) as channel compensation method. In contrast to the joint factor analysis (JFA), EVA compensated emotional variability as the channel component on its supervector representation. Furthermore, the i-vector technique was used to estimate and model the speakers. The deep discriminant analysis (DDA) was used as a channel compensation method. The effectiveness of the EVA-based i-vector was evaluated in the speaker verification task of the SUSAS dataset with three different scoring methods. Compared to the standard i-vector system, EVA-based i-vector represented the speaker's supervector more effectively by presenting a different sub-cluster for different emotions. A better result has been achieved with the lowest equal error rate (EER) of 4.51%, 4.37%, and 4.08% for Cosine similarity scoring (CSS), Euclidean distance scoring (EDS), and Mahalanobis distance scoring (MDS), respectively. In the ablation experiment, the number of mixture GMM-UBM systems and the number of the EVA dimensions have been explored. The result shows that the number of mixture GMM-UBM system and the number of EVA dimensions affected the model performance. A 512-mixture GMM-UBM system obtained the best performance with EER 4.07%, and the number of EVA dimensions is 600-dimensional with EER 4.11%.

As mentioned in Section 3.1.3, EVA-based i-vector use total-variability factor $w$ to model the speaker in a single-rank representation. It means the speaker and emotion variability is modeled by a single matrix $T$. It resulted in the speaker and emotion subspace does not model as well as the eigenvoice matrix $V$. Therefore, the first interesting future work is to estimate a $T_{new}$ matrix (different from the original $T$ matrix), which better represents the speaker and emotion space. It is performed in two steps. The first step is to train the $V$ matrix exactly as matrix $T$ by assuming the given segments belong to a single class. The second step is to initial $T_{new}$ matrix using the $V$ matrix. In this step, each segment is trained as belonging to different classes.

Moreover, the use of a fixed threshold makes the speech that has different emotions get the same treatment. Therefore, the interest direction of future work is to explore the dynamic thresholding that adaptive to different emotional conditions. Thus, the second interesting future work is to incorporate an algorithm that modulates the threshold adaptively for resulting in a time-adaptive thresholding $\eta_t$, where $t$ is time.

**Author Contributions:** Theory and conceptualization, B.H.P. and H.T.; data requirement, B.H.P. and H.T.; methodology, B.H.P. and H.T.; software design and development, B.H.P.; validation, B.H.P., H.T. and K.T.; formal analysis, B.H.P. and H.T.; investigation, B.H.P.; writing–original draft preparation, B.H.P.; writing–review

and editing, B.H.P., H.T. and K.T.; visualization, B.H.P. and H.T.; supervision, H.T. and K.T. All authors have read and agreed to the published version of the manuscript.

**Funding:** This research received no external funding.

**Acknowledgments:** I would like to thank Tamura Laboratory that supported us in this works. Thank you to LDC for allowing us access to the SUSAS database.

**Conflicts of Interest:** All authors have no conflict of interest to report.

## Abbreviations

The following abbreviations are used in this manuscript:

| | |
|---|---|
| CSS | Cosine Similarity Scoring |
| DDA | Deep Discriminant Analysis |
| EDS | Euclidean Distance Scoring |
| EER | Equal Error Rate |
| EVA | Emotional Variability Analysis |
| GMM | Gaussian Mixture Model |
| JFA | Joint Factor Analysis |
| KL | Kullback–Leibler |
| LDA | Linear Discriminant Analysis |
| LDC | Linguistic Data Consortium |
| LPCC | Linear Predictive Cepstral Coefficients |
| MAE | Mean Absolute Error |
| MDS | Mahalanobis Distance Scoring |
| MDE | Minimum Divergence Estimation |
| MSE | Mean Squared Error |
| MFCC | Mel-frequency Cepstral Coefficients |
| PLDA | Probabilistic Linear Discriminant Analysis |
| ReLU | Rectified Linear Activation Unit |
| SAD | Speech Activity Analysis |
| SGD | Stochastic Gradient Descent |
| SUSAS | Speech Under Simulated and Actual Stress |
| t-SNE | t-distributed Stochastic Neighbor Embedding |
| TVM | Total-Variability Model |
| UBM | Universal Background Model |

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
