# Peer review of "Emotional Variability Analysis Based I-Vector for Speaker Verification in Under-Stress Conditions"

_electronics, doi:10.3390/electronics9091420_

Round 1

Reviewer 1 Report

This paper proposes a speaker modelling that accommodates the presence of emotions on the speech segments by extracting a speaker representation compactly. The paper is well written and organise, however there are some typos and minor grammatical mistakes.

The authors consider other baseline methods to compare the results along with other publicly available datasets. What is the main objective of this study? What are the contributions? The authors should also consider other loss functions. 

Reviewer 2 Report

This paper present a speaker modeling that accommodates the presence of emotions on the speech segments by extracting a speaker representation.  The proposed model was built based on i-vector and used deep discriminant analysis as compensation.  Compared with standard i-vector/DDA, the proposed method shows a reduction of error rate from ~6% to ~4%.   

Although there is concern about influence of this paper over the readers of Electronics, the methods and results in this manuscript is well described and presented.  The overall proposed organization and section structure are consistent with the proposed objectives.  To further improve the integrity of this work, it would be good to discuss the limitation of this current work and how to overcome these barriers in future development.
